# Geomagnetic Field (GMF)-Dependent Modulation of Iron-Sulfur Interplay in *Arabidopsis thaliana*

**DOI:** 10.3390/ijms221810166

**Published:** 2021-09-21

**Authors:** Gianpiero Vigani, Monirul Islam, Viviana Cavallaro, Fabio F. Nocito, Massimo E. Maffei

**Affiliations:** 1Department of Life Sciences and Systems Biology, Plant Physiology Unit, University of Turin, Via Quarello 15/a, 10135 Turin, Italy; monirul.islam@unicatt.it (M.I.); massimo.maffei@unito.it (M.E.M.); 2Dipartimento di Scienze Agrarie e Ambientali—Produzione, Territorio, Agroenergia, Università degli Studi di Milano, 20133 Milano, Italy; viviana.cavallaro@unimi.it (V.C.); fabio.nocito@unimi.it (F.F.N.)

**Keywords:** geomagnetic field, iron, plant nutrition, sulfur

## Abstract

The geomagnetic field (GMF) is an environmental factor affecting the mineral nutrient uptake of plants and a contributing factor for efficient iron (Fe) uptake in Arabidopsis seedlings. Understanding the mechanisms underlining the impact of the environment on nutrient homeostasis in plants requires disentangling the complex interactions occurring among nutrients. In this study we investigated the effect of GMF on the interplay between iron (Fe) and sulfur (S) by exposing *Arabidopsis thaliana* plants grown under single or combined Fe and S deficiency, to near-null magnetic field (NNMF) conditions. Mineral analysis was performed by ICP-MS and capillary electrophoresis, whereas the expression of several genes involved in Fe and S metabolism and transport was assayed by qRT-PCR. The results show that NNMF differentially affects (i) the expression of some Fe- and S-responsive genes and (ii) the concentration of metals in plants, when compared with GMF. In particular, we observed that Cu content alteration in plant roots depends on the simultaneous variation of nutrient availability (Fe and S) and MF intensity (GMF and NNMF). Under S deficiency, NNMF-exposed plants displayed variations of Cu uptake, as revealed by the expression of the *SPL7* and *miR408* genes, indicating that S availability is an important factor in maintaining Cu homeostasis under different MF intensities. Overall, our work suggests that the alteration of metal homeostasis induced by Fe and/or S deficiency in reduced GMF conditions impacts the ability of plants to grow and develop.

## 1. Introduction

Mineral nutrients are essential resources for plant growth and development. Therefore, understanding the mechanisms underlying the ability of plants to efficiently acquire nutrients from the soil and distribute them among the organs represents a scientific priority [1]. Several environmental factors (abiotic and biotic) affect plant mineral nutrition, and, conversely, maintaining the optimal nutrient status allows the plant to respond efficiently to the surrounding environment.

Understanding the mechanisms underlining the impact of the environment on nutrient homeostasis in plants requires disentangling the complex interactions occurring among nutrients. In this context, iron (Fe) availability is a good example. Indeed, Fe is present in massive amounts in soils, being the fourth most abundant element in the Earth’s crust in percentage after oxygen, silicon, and aluminum. Therefore, the widespread limited availability of Fe for plant nutrition is not related to its content in the soil but rather to its limited solubility. Low Fe availability occurs typically in calcareous soils, where Fe displays an extremely low solubility. Moreover, Fe homeostasis displays complex and not fully characterized interactions with other nutrients, such as sulfur (S). 

Sulfur is essential for optimal plant growth and development, including crop productivity. Its effects have been traditionally ascribed to its role in mitigating stressful conditions in plants and, more importantly, integrating the uptake and metabolic pathways of other nutrients. Indeed, S interacts with some macronutrients (N, P, and K) and micronutrients (Fe, Mo, Cu, Zn, and B). Such interactions could be related to different factors, such as that: (i) S shares similar chemical properties with another element (e.g., Mo or Se), competing therefore for acquisition/transport process (SULTR transporters); (ii) some metabolic and molecular processes regulating plant responses to nutritional deficiencies require the simultaneous presence of S with other nutrients (S-containing metabolites are the precursor for the synthesis of ethylene and phytosiderophores); (iii) S directly interacts with other elements (e.g., Fe) by forming complexes and chemical bonds, such as in Fe-S clusters.

Several pieces of evidence highlight the interplay occurring between Fe and S in plants. For example, it has been reported that S deficiency inhibits the response to Fe deficiency by modulating the expression of genes encoding Fe-regulatory proteins in *Arabidopsis thaliana* [2]. Furthermore, in tomato (*Solanum lycopersicum* L.) and rapeseed (*Brassica napus* L.), S deficiency limits the expression and activity of Fe-deficiency response genes under simultaneous Fe starvation [3,4]. Similarly, in barley (*Hordeum vulgare* L.) seedlings, S supply restored Fe uptake under Fe deficiency [5]. Additionally, Fe deficiency triggers responses associated with S deficiency in tomato and durum wheat (*Triticum turgidum* L. subsp. *durum*) [6] and references cited therein.

It has been recently reported that, among the environmental factors affecting plant nutritional status, the geomagnetic field (GMF) may also influence plants’ ability to acquire nutrients [7]. The GMF is a natural component of our environment [8]. The strength of the GMF at the surface of the Earth ranges from less than 30 μT in an area that includes most of South America and South Africa (the so-called South Atlantic Anomaly) to over 60 μT around the magnetic poles in northern Canada, the south of Australia, and in parts of Siberia [9]. Plants respond to both the inclination [10] and intensity [11] of the magnetic field. However, specific experimental conditions useful to decipher the effect of geophysical fields such as the GMF on plants are difficult to realize. To study the effect of GMF on plants, it is possible to expose plants in conditions where the GMF is altered by either increasing or reducing the magnetic field (MF) flux [8]. Such a condition is achievable using GMF compensation systems such as the one we developed using a triaxial Helmholtz coil (three computer-controlled orthogonal Helmholtz coils) [12].

Recently, the reduction of the GMF to near null MF (NNMF) conditions was found to affect the nutrient status of plants [7,13,14]. In addition, specific evidence has also been provided about the impact of GMF on metal homeostasis in plants. Notably, under NNMF, Fe homeostasis is influenced, and Fe-deficiency responsive genes are affected during the early stage of development. However, progress is needed to identify the molecular mechanisms responsible for such effects [7].

This work investigated the role of the GMF in the Fe/S interplay in Arabidopsis.

## 2. Results

### 2.1. GMF Differentially Affects Plant Growth under S and Fe Deficiency

Root length (RL) and shoot area (SA) were selected as morphometric parameters to monitor plant growth under different nutritional conditions in GMF- and NNMF-exposed plants. Two-way ANOVA revealed that both nutrients availability (NA) and MF intensity significantly affected RL and SA. However, only SA displayed a significant interaction (*p* = 0.017) between variables (NAxMF) (Appendix A). A pair comparison analysis (*t*-test) revealed that under control (C) and -Fe conditions, both root length and shoot area were affected in NNMF, with respect to GMF (Figure 1). However, under NNMF condition, S deficiency slightly affected plant growth compared with C and -Fe conditions. Indeed, under S deficiency (regardless of the Fe provision), NNMF affected only root length, while shoot area was not affected (Figure 1). Plants exposed to NNMF showed a decrease in total Fe contents only under the C condition, as compared to GMF-exposed plants (*p* < 0.05), while under single or combined Fe and S deficiency, the NNMF condition did not affect the total Fe content (*p* > 0.05). To monitor the S nutritional status, the indicator of S deficiency (S index) was determined according to [15]. The two-way ANOVA revealed that the MF variable did not affect the S index (*p* = 0.734), while NA significantly impaired it (*p* < 0.001). Moreover, a significant interaction among variables (NAxMF) was observed (*p* = 0.003). As expected, the S index strongly increased under -S conditions (both -S and -FeS) (its increase means that an S deficiency occurs in tissues) regardless of GMF variation. However, under –S, NNMF-exposed plants significantly decreased in such indicators with respect to GMF-exposed plants (*p* < 0.05; Figure 1). Such a variation is attributable to the change in the PO_4_^3−^ content: under –S, NNMF exposed plants displayed a decrease in PO_4_^3−^ content compared to GMF conditions (Appendix A). On the other hand, no significant (*p* < 0.05) variation occurred in Cl^−^ and NO_3_^−^ accumulation in different growth media under GMF or NNMF growing plants (Appendix A). The variation in PO_4_^3−^ content is in agreement with the expression of *PHT1* (phosphate transporter 1) in root tissues: NNMF-exposed plants displayed a downregulation (*p* < 0.05) of *PHT1* only under –S condition, as compared to GMF-exposed plants (Appendix A).

### 2.2. GMF Affects the Fe/S Interplay as Revealed by the Expression of Fe- and S-Responsive Genes

To investigate the effect of GMF on the Fe and S interplay, the expression of some Fe- and S-deficiency-responsive genes were investigated in the roots of plants exposed to GMF or NNMF and grown under C, -Fe, -S, and -FeS conditions. 

Fe-deficiency responsive genes considered were *IRT1* (iron-regulated transporter 1), *AHA2* (plasma membrane proton ATPase 2); *FRO2* (ferric reduction oxidase 2), *PYE* (Popeye, *bHLH47*), *BTS* (Brutus), *FIT*, *bHLH38* and *bHLH39* (FIT, Fe-deficiency induced transcription factor/bHLH) (Figure 2). The expression of these genes was significantly affected by the NA in the media (*p* < 0.05), while a significant interaction (NAxMF) among variables was observed for the expression of *IRT1* (*p* < 0.001), *FRO2* (*p* < 0.001), *bHLH38* (*p* < 0.001), *PYE* (*p* < 0.001) and *BTS* (*p* < 0.05) (see Appendix A for statistics).

The effect of GMF on Fe uptake-related genes was not relevant in plants grown under C and -S, while the expression of some genes was affected by GMF under Fe deficiency (–Fe and –FeS) conditions (Figure 2). Notably, under –Fe, the expression of *PYE* and *BTS* was induced to a lower extent in NNMF-exposed plants, as compared to GMF-exposed plants. Accordingly, an induction of *IRT1*, *AHA2*, *FRO2* and *FIT* (even though not significant *p* > 0.05) was detectable in NNMF-exposed plants, as compared with GMF-exposed plants under -Fe (Figure 2). Although GMF reduction did not affect the expression of Fe-uptake related genes under -S, the combined -Fe-S deficiency revealed that a higher upregulation of *IRT1*, *FRO2* and *bHLH38* occurred in NNMF- with respect to GMF-exposed plants (Figure 2). Overall, such results indicate that GMF differentially affects the expression of Fe uptake genes under –Fe and the simultaneous alteration of Fe and S availability.

The S-deficiency responsive genes considered are the following sulfate transporter genes: *SULTR 1;1* (sulfate transporter 1;1), *SULTR 1;2* (sulfate transporter 1;2), *SULTR 1;3* (sulfate transporter 1;3), *SULTR 2;1* (sulfate transporter 2;1), *SULTR 2;2* (sulfate transporter 2;2) and sulfur assimilation key enzyme genes: *APR1* (APS reductase 1), and *APR2* (APS reductase 2) (Figure 3). The two-way ANOVA revealed a significant interaction among variables (NAxMF) for the expression of the sulfate transporter genes (*SULTR1;1*, *SULTR1;2*, *SULTR1;3*, *SULTR2;1*; *SULTR 2;2*) (*p* < 0.001), while such an interaction was not observed for the expression of *APR1* and *APR2* (*p* > 0.05) (Appendix A). A paired comparison analysis revealed that under the control condition, NNMF-exposed plants showed a lower transcript level of *SULTR1;2*, *APR1*, and *APR2*, and a higher transcript level of *SULTR1;3* compared to the GMF-exposed plants. Similar results were observed under the –Fe condition, where a lower transcript accumulation of *SULTR 1;2*, *APR1* and *APR2* occurred in NNMF-exposed plants (Figure 3). Under the –S condition, plants exposed to NNMF showed a significantly lower accumulation of the S responsive genes (*SULTR 1;1*, *1;3*, *2;1*, *2;2*) (*p* < 0.05), *APR1*, and *APR2* (*p* < 0.05), but not for *SULTR 1;2* (*p* > 0.05), when compared to GMF exposed plants (Figure 3). In particular, under GMF, the expression of *SULTR1;1* displayed a 24-fold and 50-fold increase in the –S and –Fe-S plants, respectively, with respect to C. Under NNMF, the expression of *SULTR1;1* displayed a 9-fold and 20-fold increase in –S and –Fe-S, respectively, with respect to the C (Figure 3). Furthermore, under combined Fe and S deficiency (-Fe-S), a significantly lower induction (*p* < 0.05) in the NNMF- as compared to the GMF-exposed plants, was found only for the expressions of *SULTR 1;1*, *APR1*, and *APR2* (Figure 3).

### 2.3. GMF Affects the Impact of Fe and S Availability on Metal Accumulation in Plant Tissues

The content of metals such as Fe, Mn, Cu, Zn, and Mo, was determined in both the root and shoot tissues of the NNMF- and GMF-exposed plants grown under the different conditions of nutrient availability (C, -Fe, -S, -Fe-S). At the root level, two-way ANOVA revealed that (i) NA significantly affected Fe (*p* < 0.001), Zn (*p* < 0.001), and Mo (*p* < 0.001) content; (ii) MF significantly affected Cu (*p* < 0.001) and Zn (*p* < 0.05); (iii) the NAxMF interaction significantly affected only Cu content (Table 1). In particular, under control conditions, NNMF-exposed plants displayed low Fe content compared to GMF- exposed plants, while the content of other metals did not change, in agreement with our previous findings [14]. Under –Fe, the content of Fe, Mn, Cu, and Zn in the roots were higher in the NNMF- than in the GMF-exposed plants (*p* < 0.05; Table 1). Under -S conditions, only a change in Cu content (*p* < 0.05) was observed in the roots of NNMF- relative to GMF-exposed plants. Under the combined –Fe-S deficiency, NNMF-exposed plants displayed a significantly (*p* < 0.05) higher content of Mn, Fe, Cu, and Zn, compared to GMF conditions (Table 1). At the shoot level, two-way ANOVA revealed that NA significantly affected Fe (*p* < 0.001), Cu (*p* < 0.001), Zn (*p* < 0.001), and Mo (*p* < 0.001) content, while MF and NAxMF did not affect metal contents. However, under the –Fe condition, a significant (*p* < 0.05) variation in Fe content was observed in NNMF- when compared to GMF-exposed plants, while the content of Mn, Cu, Zn, and Mo was not affected by NNMF under C, -Fe, and –FeS conditions (Table 1).

### 2.4. GMF Affects Cu Homeostasis Depending on the Plant Nutritional Status

Since we previously identified a link between Fe and Cu under NNMF conditions [14], we monitored the expression of *SPL7* and *miR408*, two master regulator genes involved in the cross-link between Fe and Cu homeostasis in plants [16,17]. Two-way ANOVA revealed that NA (*p* < 0.001), MF (*p* < 0.001), and NAxMF (*p* < 0.05) affected *SPL7* expression (Figure 4). Accordingly, *miR408* expression was significantly affected by MF (*p* < 0.001), NA (*p* < 0.001) as well as NAxMF interaction (*p* < 0.001) (Appendix A). In particular, under S deficiency (both –S and –Fe-S) conditions, NNMF-exposed plants displayed downregulation of *SPL7* compared to GMF condition (*p* < 0.05). By contrast, NNMF-exposed plants showed a downregulation of *miR408* under C, -Fe and –Fe-S. Similar *miR408* transcript accumulations were observed under –S in NNMF- and GMF-exposed plants (Figure 4).

In order to provide further evidence about the effect of GMF on Cu homeostasis, we monitored the growth of: (i) transgenic lined defective in *SPL7* and (ii) transgenic knockdown lines transformed with artificial microRNA (amiRNA) constructs targeting either *FRO5* or both *FRO4* and *FRO5* (the levels of *FRO4* and *FRO5* transcripts were reduced), according to [16]. *FRO4* and *FRO5* encode for root surface Cu(II) chelate reductases that are transcriptionally upregulated dependent on *SPL7* in response to Cu deficiency [14]. Our results revealed that under NNMF conditions, the metal profile of *spl7* mutants showed lower Fe contents and higher Cu and Mn contents compared to GMF (Table 2).

Furthermore, the whole seedlings of wt Col0 showed low Fe contents and a trend of higher accumulation of Cu and Zn under NNMF conditions, as compared to GMF conditions. On the other hand, *ami4/5* showed high contents of Mn and lower Cu contents under NNMF conditions, as compared to GMF conditions. Similarly, *ami5* showed a higher Fe and Cu content accumulation under NNMF conditions, as compared to GMF conditions. Furthermore, the combined effect of MF and NA on the root growths of the *spl7*, *amiFRO5*, *amiFRO4/5* lines were also monitored (Figure 5). While NNMF-Col0 exposed plants displayed a decreased RL under all the nutrient conditions (C, -Fe, -S, -Fe-S), NNMF- *spl7* exposed plants displayed an RL decrease only under the –Fe-S condition, as compared with GMF (Figure 5). On the other hand, both *ami4/5* and *ami5* did not show root length changes under NNMF in all nutrient conditions (Figure 5).

## 3. Discussion

In this work we provide further evidence on the effect of GMF on plant mineral nutrition by characterizing the Fe/S interactions in plants growing under NNMF.

Our results show that NNMF affects Fe content and Fe homeostasis, in agreement with our previous observations [7]. The impact of NNMF a few days after germination highlights that Fe uptake genes are upregulated compared to GMF under control conditions, with a lower induction under Fe deficiency. Some Fe-responsive genes were affected only under –Fe and –Fe-S conditions. In particular, under Fe deficiency, in NNMF exposed plants the expression of *AHA2*, *PYE*, and *BTS* decreased with respect to the GMF condition, while the expressions of *IRT1*, *FRO2* and *bHLH38* were affected by NNMF only under a combined Fe and S deficiency. Such observations indicate that GMF is a contributing factor to proper Fe homeostasis in plants. The difference observed in our previous observation [7] might be explained considering that several GMF-responsive genes show biphasic dose-dependent expression, indicating a hormetic response of Arabidopsis to MFs [11]. The different contents of Fe and other metals in the root and shoot tissues of plants exposed to NNMF under the nutritional conditions might explain the GMF-dependent modulation of Fe-responsive genes. Under Fe deficiency, the higher Fe content in NNMF-exposed, as compared to GMF-exposed, plants suggests a less severe Fe deficiency perceived by plants under NNMF, affecting the induction of the Fe uptake system. However, the combined Fe and S deficiency induced the Fe uptake system under NNMF, suggesting that GMF has an impact on the regulation of the Fe/S interaction.

GMF reduction affects the expression of S-responsive genes, as well as the content of SO_4_^2−^. Different indicators of S nutrition based on plant analysis have been proposed. However, Sorin et al. [15] and Etienne et al. [18] proposed and validated the ([Cl^−^] + [NO_3_^−^] + [PO_4_^3−^]):[SO_4_^2−^] ratio as a relevant indicator of S deficiency in plant species. This ratio is related to the anion balance regulation occurring in plants. Sulfur availability affects the inorganic anion balance in plants, involving Cl^−^, NO_3_^−^, and PO_4_^3−^. Sulfate also acts as an osmoticum in the cell, and its remobilization during S deprivation is compensated osmotically by a vacuolar accumulation of Cl^−^, NO_3_^−^, and PO_4_^3−^. In our experiments, GMF affects the S index under S deficiency.

On the other hand, Narayana et al. [13] demonstrated that the anion content (such as NO_3_^−^, SO_4_^2−^, and PO_4_^3−^) and the gene expression of anion transporters and channels are affected early, a few days after NNMF exposure. Our results suggest that after seven days of NNMF exposure, the anion balance is affected mainly under S deficiency, suggesting a link between GMF perception and the S-nutritional status of plants. Such variation is attributable to the decrease of PO_4_^3−^ content. Accordingly, the combined effect of MF and S deficiency affects the expression of phosphate transporter *PHT1* at the root level (Appendix A). 

The existence of a coordination between S and P homeostasis in plants has been proposed [19,20]. While S-deficient plants display a rapid replacement of sulfolipids by phospholipids, the replacement of phospholipids by sulfolipids occurs in Pi deficient plants [21,22,23]. Such a metabolic switch confirms the P/S nutritional interdependency. Although no direct evidence supporting the shift between sulfolipids and phospholipids under NNMF conditions is available so far, we recently demonstrated that GMF also affects plant lipid composition [14]. While the regulation of lipid composition is a complex process, a link between the variation of lipid composition and the modulation of H^+^ -ATPase activity has been suggested. Oh et al. [24] demonstrated that cytochrome B5 reductase 1 (CBR1) could activate plasma membrane-localized H ^+^ -ATPases by facilitating the accumulation of unsaturated fatty acids. Indeed, CBR1 plays a role in activating fatty acids desaturase 2 (FAD2) and fatty acids desaturase 3 (FAD3), allowing for double bonds into fatty acids. GMF affects both the expression of *AHA2* and the fatty acid composition. While NNMF strongly induced *AHA2* expression a few days after germination [7], this work demonstrates that seven days of NNMF exposure revealed a decrease of *AHA2* under Fe deficiency at the root level. Such results might reflect the biphasic gene expression of plants exposed to MFs, as recently highlighted by Paponov et al. [11]. On the other hand, we recently observed that several fatty acids (FAs) were affected by the reduction of the GMF, including palmitic (C16:0), palmitoleic (C16:1), stearic (C18:0), oleic (C18:1), linoleic (C18:2), and linolenic (C18:3) acids [14].

Evidence for the co-regulation of P and S signaling pathways is starting to emerge. Rouached et al. [25] suggested that an adaptive regulation of the SO_4_^2−^ transport and distribution process occurs under Pi deficiency in plants. Here we provide evidence that S-deficient plants displayed both a low PO_4_^3−^ content and a decreased expression of some SULTR gene (*SULTR1;1*, *1;3*, *2;1*, *2;2*) under th NNMF condition. Under combined Fe and S deficiency, variations of PO_4_^3−^ content and *SULTR* expression were unaffected under the NNMF condition, as compared with the GMF condition, suggesting that a low Fe availability does not affect PO_4_^3−^ content or, in turn, the regulation of SULTR genes, except for SULTR1;1. This observation indicates that (i) Fe presence is an essential factor for the S/P interplay under NNMF conditions, and that (ii) other pathways are involved in the GMF-dependent regulation of the P/S interaction in plants. Considering the high level of complexity and interconnection in the regulation of SO_4_^2−^and Pi homeostasis in plants, future studies are underway to disentangle such interactions in response to the GMF.

Our results showed that both Fe and S deficiency induced Fe and S-responsive genes, respectively, indicating that plants perceived such deficiencies in our experimental conditions. The effect of S deprivation on Fe deficiency symptoms could be due to a differential accumulation of non-iron metals like Mn and Zn [2]. In Arabidopsis, the negative regulation (effect) of S deficiency on the Fe uptake machinery induced in response to Fe deficiency can be explained as a need of plants to limit the unspecific transport of potentially toxic divalent cations (Mn and Zn) into the roots through the activity of IRT1 and Nramp1 [2]. Here, we partially observed an effect of –S on Fe deficiency-induced genes: Fe/S interaction could occur in some species or developmental stage with specific mechanisms. Indeed, it could not be excluded that growth parameters may also influence these responses [2]. However, in this work, we observed that GMF differentially affects Mn, Zn, and Cu in the root and leaf tissues, and that MF intensity alters the different impact on the metal content in plants grown under different Fe and S availabilities. 

The Cu content alteration in roots depends on the simultaneous variation of nutrient availability (Fe and S) and MF intensity (GMF and NNMF). A link between GMF and Fe-Cu crosstalk has been reported recently [7]. Accordingly, the expression of *At5g52680*, a gene encoding a Cu transport protein, increased in the root of Arabidopsis plants exposed to NNMF [11]. Here we confirm this link, and observe that GMF also affects Cu homeostasis under S availability. To investigate how Cu homeostasis is modulated under the different conditions considered, we focused our analysis on some molecular factors involved in plant Cu-Fe crosstalk, such as SPL7. It has been revealed that SPL7 acts as a repressor of some aspects of Fe uptake [16]. Notably, SPL7 represses (directly or indirectly) the expression of some Fe-responsive genes under Cu deficiency and combined Fe and Cu deficiency [24]. SPL7 binds to Cu-responsive elements within the miR408 promoter and activates its expression [26]. Here we showed that S availability affects *SPL7* expression in plants exposed to NNMF. Accordingly, S deficiency leads to a downregulation of *miR408* both under GMF and NNMF conditions. As all *miR408* targets are mRNAs that encode apoplastic cuproproteins, a role of *miR408* in Cu redistribution has been postulated under Cu depletion [3]. Such results suggest that S availability is an important factor in maintaining Cu homeostasis under different MF intensities. A possible regulating pathway responsible for both S and Cu responses involves *elongated hypocotyl 5* (*HY5*), which is important for regulating plant growth and a master regulator of root morphogenesis [27]. Previous studies showed that *HY5* plays a role in the regulation of *APR* (sulfate assimilation) [28] and *SPL7* gene expression [3]. Our data revealed that GMF affects *HY5* expression differentially under single or combined Fe and S deficiency (Appendix A), except for single S deficiency. Accordingly, S-deficient plants did not display RL variation under NNMF conditions. However, under the control and Fe deficiency (single or combined) conditions, *HY5* expression decreased under NNMF, suggesting that the HY5-mediated pathway might impair the GMF-dependent modulation of nutrient uptake at the root level. These data are in agreement with our previous observations [29]. Although evidence about the involvement of ROS signaling in the HY5-mediated regulation of RSA has been provided, the mechanism by which HY5 regulates root photomorphogenesis is mainly unknown [27].

## 4. Materials and Methods

### 4.1. Plant Materials, Media Composition and Growing Conditions

The *Arabidopsis thaliana* ecotype Columbia-0 (Col-0), wild type (WT), and the *spl7*, *amiFRO5*, and *amiFRO4/5* mutant lines (kindly provided by Prof Ute Kramer) were used for this study. For agar plate cultivation, seeds were surface sterilized using 70% ethanol (*v/v*) for 2 min and 5% (*w/v*) calcium hypochlorite for 5 min. After multiple washes with sterile double distilled water, seeds were sown on the surface of sterile agar plates (12 × 12 cm) following the medium composition of Gruber et al. [30] (Appendix A). Plants were grown in the presence (C) or the absence (-Fe) of Fe, in the absence of S (-S), or in a combination of Fe and S deficiency (-Fe-S) (Appendix A). Plates were sealed with Micropore tape to allow gas exchange and avoid condensation, and kept in the dark at 4 °C for 48 h for synchronizing germination Finally, plates were transferred to GMF (control) or NNMF (treatment) at the same time and in the same laboratory, and provided with white light 130 μmol m^−2^ s^−1^ by a high-pressure sodium lamp source (SYLVANIA, Grolux 600W, Bruxelles, Belgium). A blue spotlight film was used to reduce the red component of the lamps. Plants were grown under 16 h/8 h light and dark cycles at 22 °C (±1.5 °C). All experiments were performed under normal gravity.

### 4.2. GMF Reduction System and Plant Exposure

The local GMF values, typical of the Northern hemisphere at 45°0′5″ N and 7°36′58″ E coordinates, ranged between 41 to 43 µT. To generate the NNMF condition, orthogonal Helmholtz coil systems were used as previously detailed [12], ranging between 40 to 44 nT. According to a previous investigation [7], the reduction of GMF to NNMF affects the accumulation of metals in plant tissues and the Fe uptake machinery genes. Therefore, to unravel the interplay between Fe and S under NNMF conditions, plates containing different media with Arabidopsis seeds were exposed either to GMF or NNMF for 1 week. Similarly, double-blinded experiments were performed to evaluate control and NNMF treatments.

### 4.3. Morphological Measurement

After 1 week of exposure, Petri dishe pictures were taken to measure root length and shoot area using ImageJ software [31]. 

### 4.4. Metals Analysis in Plant Tissues 

Tissues (shoots, roots or whole plants) of plants exposed to either GMF or NNMF were washed with Milli Q water and dried in a ventilated oven at 70 °C for 4 days. The root and shoot dry weights were measured, and tissues were then digested with 500 µL 65% HNO_3_ for 4 h at 120 °C and later at 200 °C to fully evaporate HNO_3_. Once mineralized, 250 µL of 65% HNO_3_ was added, and vortexed samples were transferred into polypropylene test tubes with a 1:40 dilution with Milli-Q water. Finally, the mineral contents of the samples were measured by inductively coupled plasma-mass spectrometry ICP-MS (BRUKER Aurora- M90 ICP-MS), as previously described [7]. Analyses were carried out from four independent biological replicates.

### 4.5. Anions Analysis by Capillary Electrophoresis (CE)

To measure total anion contents, Arabidopsis whole tissues exposed 1 week to GMF or NNMF as described above were collected and washed with Milli-Q water, soft soaked with paper, immediately frozen in liquid nitrogen, and stored at −80 °C. Anions were then extracted with ultrapure water using a mortar and pestle. The extracts were filtered using 0.22 µm filters and assayed by capillary electrophoresis (Agilent 7100, Agilent Technologies, Santa Clara, USA). Anions such as chloride (Cl^−^), sulfate (SO_4_^2−^), nitrate (NO_3_^−^) and phosphate (PO_4_^3−^) were analyzed through a bare fused silica capillary column with extended light path BF3 (i.d. = 50 μm, I = 72 cm, L = 80.5 cm). Sample injection was followed by 50 mbar pressure for 4 s with −30 kV voltage and detection at the 350/380 nm wavelength. All anions were identified by using pure standards. The final anion content in each sample was calculated as µg g ^−1^ fresh weight (f wt). Experiments were repeated at least 3 times.

### 4.6. RNA Purification from Arabidopsis Root Tissues

For transcriptomic analyses, *A. thaliana* WT roots were collected after 1-week growth under GMF or NNMF conditions and immediately frozen in liquid nitrogen and kept at −80 °C. Twenty milligrams of frozen plant material were ground in liquid nitrogen with mortar and pestle. The total RNA was isolated by using the Machery–Nagel RNA Isolation Mini Kit (Machery-Nagel GmbH & Com., Düren, Germany), and RNase-Free DNase, following the manufacturer’s protocols. RNA quality and quantity were checked as previously described by Islam et al. [7]. The first-strand cDNA synthesis was performed from 1 μg of total RNA using a high capacity cDNA reverse transcription kit (Applied Biosystem, Foster, USA) according to the manufacturer’s guidance, following mixture incubation at 25 °C for 10 min, 37 °C for 2 h, and 85 °C for 5 min.

### 4.7. Quantitative Real-Time PCR (qPCR)

qPCR was performed on cDNA obtained from roots of plants exposed to either GMF or NNMF. All qPCR analyses were run on a QUANTSTUDIO 3 Real-Time System (Thermo Fisher Scientific, Waltham, MA, USA) using SYBR green I with ROX as the reference dye as previously described [7]. Primers were designed using Primer 3 software (Rozen and Skaletsky, 2000), as reported in Appendix A. The thermal conditions of PCR were the following: 10 min at 95 °C, 40 cycles 15 s at 57 °C and 20 s at 72 °C. All runs were performed by a melting curve analysis from 55 to 95 °C for the genes *At4g30190, PLASMA MEMBRANE PROTON ATPASE 2 (AHA2); At1g01580, FERRIC REDUCTION OXIDASE 2; At4g19690, IRON-REGULATED TRANSPORTER 1, IRT1; At2g28160, FER-LIKE IRON DEFICIENCY INDUCED TRANSCRIPTION FACTOR, FIT; At3g56970, BASIC HELIXLOOP-HELIX 38; At3g56980, BASIC HELIX-LOOP-HELIX 39; At4g04610, APS REDUCTASE 1 (APR1); At1g62180, ADENOSINE-5′-PHOSPHOSULFATE REDUCTASE (APR2); At2g47015, MICRORNA408 (MIR408); At5g18830, SQUAMOSA PROMOTER BINDING PROTEIN-LIKE 7 (SPL7); At3g18290, BRUTUS (BTS); At3g47640, POPEYE (PYE); At5g11260, ELONGATED HYPOCOTYL 5 (HY5); At1g25540, PHYTOCHROME AND FLOWERING TIME 1 (PFT1); At4g28610, PHOSPHATE STARVATION RESPONSE 1, (PHT1); At4g08620, SULPHATE TRANSPORTER 1;1 (SULTR1;1); At1g78000, SULFATE TRANSPORTER 1;2 (SULTR1;2); At1g22150, SULFATE TRANSPORTER 1;3 (SULTR1;3); At5g10180, SULFATE TRANSPORTER 2;1 (SULTR2;1); At 1g77990,* and *SULPHATE TRANSPORTER 2;2 (SULTR2;2).* The qPCR results were normalized from GMF and NNMF using four reference genes: *At2g37620, ACTIN1 (ACT1); At5g19510, ELONGATION FACTOR 1B ALPHA SUBUNIT 2 (eEF1Balpha2); At1g13440, CYTOPLASMIC GLYCERALDEHYDE-3-PHOSPHATE DEHYDROGENASE (GAPC2);* and *At1g51710, UBIQUITIN SPECIFIC PROTEASE 6 (UBP6)*. Among the four reference genes, *eEF1Balpha2* was identified as the most stable gene using Normfinder software (MOMA, Aarhus, Denmark) [32]. Together with *eEF1Balpha2*, we also considered *UBP6,* because it was the second-most stable gene, and was previously tested by Wang et al. [33], on Fe deficiency-induced responses in Arabidopsis plants. All amplification plots were analyzed with Mx3000PTM software to obtain Ct values. The relative mRNA levels of expression of each gene were calibrated and normalized with the levels of both *eEF1Balpha2* and *UBP6* mRNA. The gene expression data were expressed to the relative expression (2^–ΔΔ^Ct) in NNMF with respect to the GMF condition. 

### 4.8. Statistical Analysis

Pair comparison analysis (*t*-test) and two-way analysis of variance (ANOVA) with mean comparisons using Tukey’s test were conducted using Past3 software. Data are means of three (qPCR analysis; *n* = 3) or four (metal content determination; *n* = 4) independent experiments each run in triplicate. Each replicate was characterized by pooled tissues (root or shoot) from 20 to 30 plants. Different letters above bars indicate statistically significant differences (*p* < 0.05).

## 5. Conclusions

The interest in investigating Fe and S interaction relies on the expected increase of dual Fe and S deficiency in soil in the coming years. It has been estimated that Fe deficiency impairs plant growth on one third of the cultivated land worldwide and concomitantly, the decrease of S release, related to anthropogenic activities, will increase S deficiency in soils [2]. Therefore, sustaining plant growth and productivity by optimizing the use of exogenous fertilizers, will necessitate a decryption of the mechanisms that govern the interconnection of plant Fe and S homeostasis. It is well known that Fe deficiency limits plant growth on one-third of the cultivated land on the the surface of the planet, while the extent of S-deficient soils is increasing. We previously demonstrated that one of the abiotic stresses experienced by plants is variation of the GMF [11]. Here we provided further evidence that: (i) GMF reduction affects nutrient homeostasis in plants; (ii) GMF has a differential impact on nutrient homeostasis in root and leaf tissues; (iii) the alteration of metal homeostasis induced by Fe and/or S deficiency in reduced GMF conditions might impact on the ability of plant growth, yield and development.

## Figures and Tables

**Figure 1 ijms-22-10166-f001:**
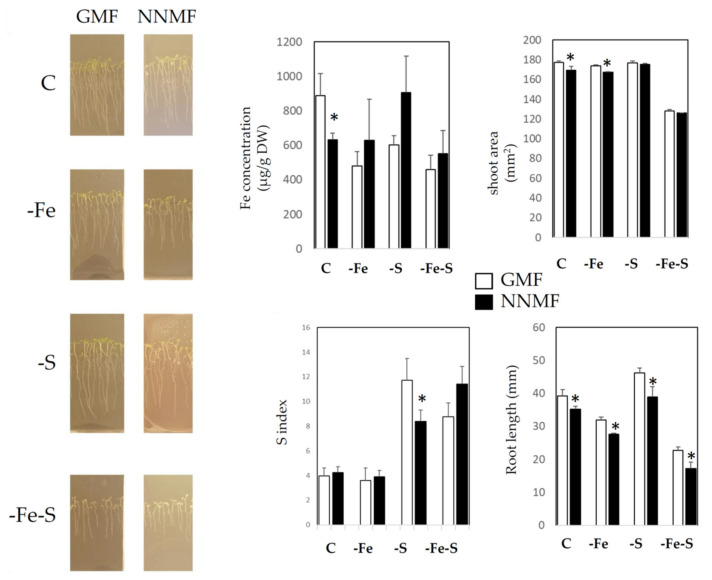
Morphometric measurements (root length and shoot area), Fe concentration, and S index of *Arabidopsis thaliana* seedlings grown under full nutrient conditions (C), absence of iron (-Fe), absence of sulfur (-S) and combined Fe and S deficiency (-Fe-S). Root length and shoot area measurements were performed 7 days after transferring seedlings from GMF to NNMF conditions. Mean values (±SE) are from 3 independent biological experiments and asterisks indicate significant (*p* < 0.05) differences between NNMF and GMF exposed plants. Two-way ANOVA results are reported in Appendix A.

**Figure 2 ijms-22-10166-f002:**
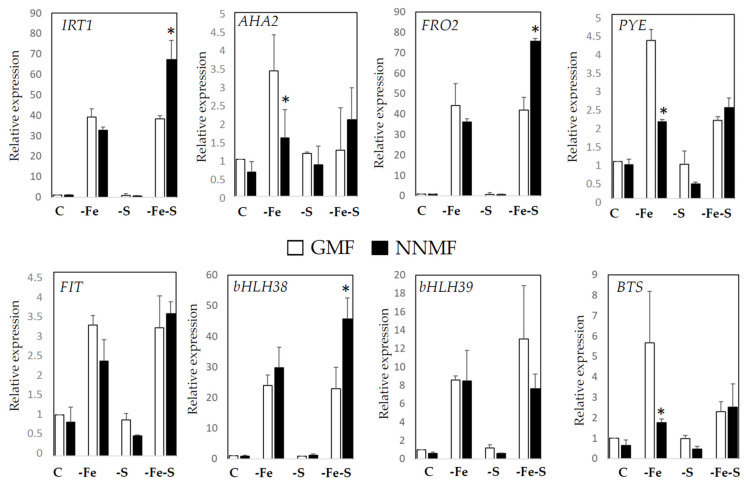
Expression of Fe deficiency-induced genes in *Arabidopsis thaliana* seedlings grown under full nutrient conditions (C), absence of iron (-Fe), absence of sulfur (-S) and combined Fe and S deficiency (-Fe-S). Plants were exposed for 7 days both to GMF and NNMF conditions. Expressions of *FRO2*, *IRT1*, *AHA2* and *PYE* genes is reported in the upper panel while expressions of *FIT*, *bHLH38*, *bHLH39* and *BTS* genes is reported in the lower panel. Data are from three independent experiments (*n* = 3). Values are expressed as fold change (±SE) with respect to control plants growing in GMF conditions under full nutrient media (C). Asterisks indicate significant (*p* < 0.05) differences between NNMF and GMF exposed plants. Two-way ANOVA results are reported in Appendix A.

**Figure 3 ijms-22-10166-f003:**
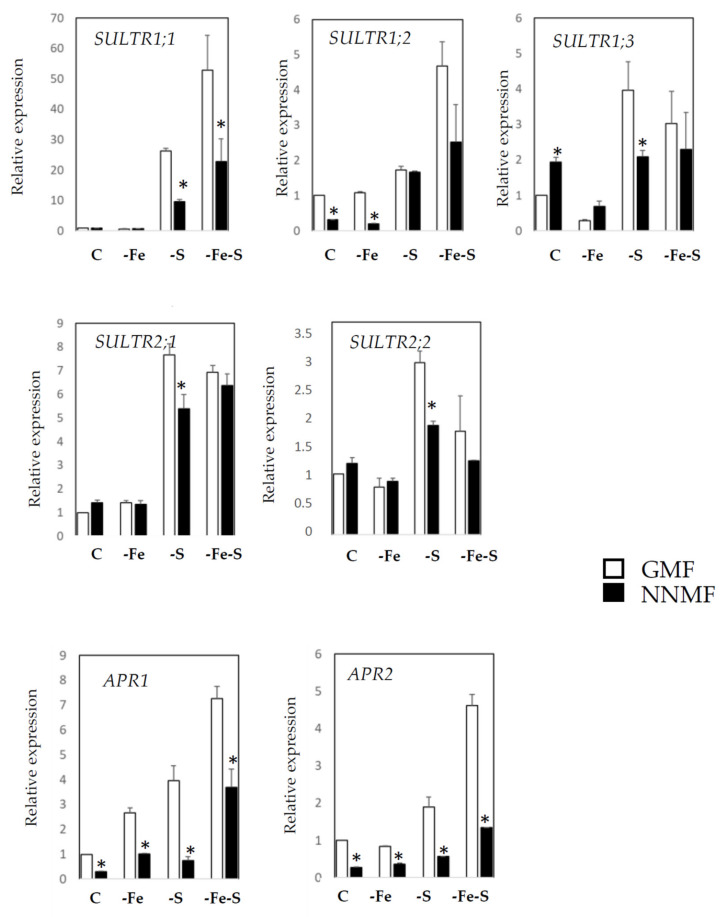
Expression of S deficiency-induced genes in *Arabidopsis thaliana* seedlings grown under full nutrient conditions (C), absence of iron (-Fe), absence of sulfur (-S) and combined Fe and S deficiency (-Fe-S). Plants were exposed for 7 days both to GMF and NNMF conditions. Expressions of *SULTR 1;1, SULTR1;2, SULTR 1;3* and *APR1* were reported in the upper panel while expressions of *SULTR 2;1, SULTR 2;2* and *APR2* are reported in the lower panel. Data are from three independent experiments (*n* = 3). Values are expressed as fold changes (±SE) with respect to control plants growing in GMF conditions under full nutrient media (C). Asterisks indicate significant (p < 0.05) differences between NNMF and GMF exposed plants. Two-way ANOVA results are reported in Appendix A.

**Figure 4 ijms-22-10166-f004:**
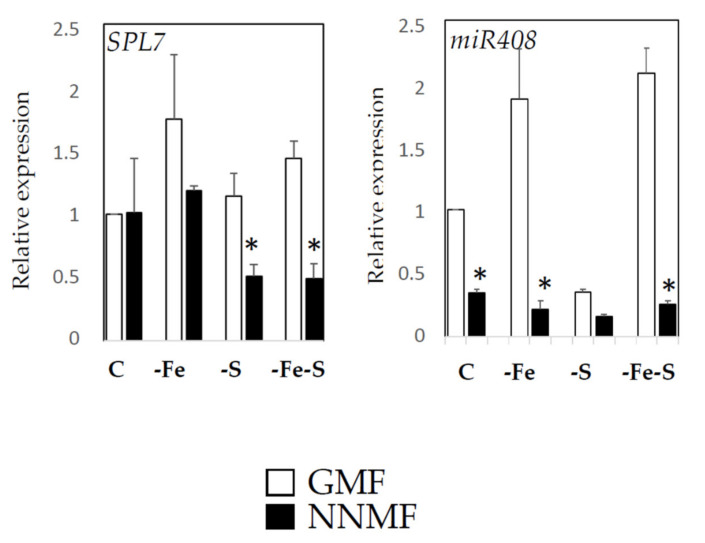
Expression of gene regulation Cu homeostasis (SPL7 and miR408) in *Arabidopsis thaliana* seedlings grown under full nutrient conditions (C), absence of iron (-Fe), absence of sulfur (-S) and combined Fe and S deficiency (-Fe-S). Plants were exposed for 7 days both to GMF and NNMF conditions. Data are from three independent experiments (*n* = 3). Values are expressed as fold changes ( ± SE) with respect to control plants growing in GMF conditions under full nutrient media (C). Asterisks indicate significant (*p* < 0.05) differences between NNMF and GMF exposed plants. Two-way ANOVA results are reported in Appendix A.

**Figure 5 ijms-22-10166-f005:**
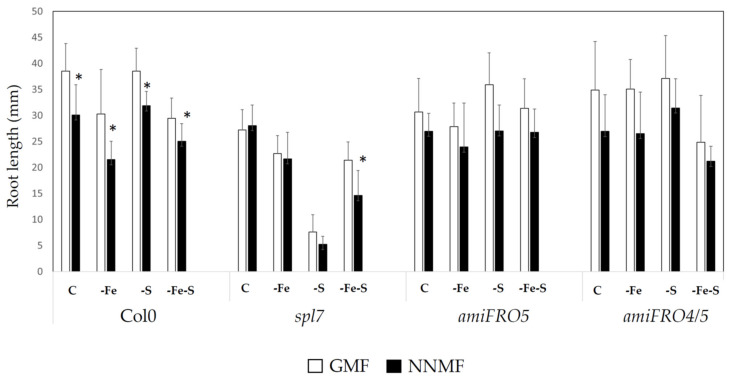
Root length of Arabidopsis thaliana lines (Col0, *spl7*, *amiFRO5*, *amiFRO4/5*) grown under full nutrient conditions (C), absence of iron (-Fe), absence of sulfur (-S), and combined Fe and S deficiency (-Fe-S). Root lengths were measured 7 days after transferring seedlings to NNMF conditions. Mean values (±SE) are from 3 independent biological experiments; asterisks indicate statistical difference (*p* < 0.05) between NNMF and GMF exposed plants.

**Table 1 ijms-22-10166-t001:** Metal contents (Fe, Zn, Cu, Mn, and Mo expressed as μg/g dry weight) of shoot and root tissues of *Arabidopsis thaliana* seedlings exposed to GMF and NNMF and grown under full nutrient conditions (C), absence of iron (-Fe), absence of sulfur (-S), and combined Fe and S deficiency (-Fe-S). Data are shown as the mean (±SE) of four independent biological experiments; asterisks indicate significant (*p* < 0.05) differences between NNMF and GMF exposed plants. Two-way ANOVA results are reported in Appendix A.

	C		-Fe		-S		-Fe-S	
ROOT	GMF	NNMF	GMF	NNMF	GMF	NNMF	GMF	NNMF
Mn	24.05 ± 13.80	68.51 ± 34.10	178.03 ± 24.95	895.90 * ± 30.70	48.15 ± 10.98	72.68 ± 24.71	138.35 ± 70.94	326.32 * ± 81.27
Fe	323.46 ± 44.21	257.38 * ± 34.78	30.68 ± 6.54	159.98 * ± 74.49	339.40 ± 25.16	300.49 ± 12.98	72.63 ± 23.32	148.84 * ± 20.26
Cu	14.50 ± 4.32	20.36 ± 3.09	8.41 ± 3.85	33.07 * ± 2.58	11.45 ± 3.73	19.31 * ± 2.73	10.31 ± 1.51	30.82 * ± 8.49
Zn	453.54 ± 203.01	552.89 ± 107.70	2427.34 ± 399.90	5213.27 * ± 607.61	538.89 ± 144.71	723.83 ± 284.09	4887.01 ± 980.71	9200.66 * ± 1502.25
Mo	7.56 ± 3.50	5.28 ± 2.61	1.01 ± 0.24	10.91 * ± 0.16	54.46 ± 10.33	55.24 ± 12.52	25.83 ± 6.41	40.80 ± 17.51
SHOOT								
Mn	13.83 ± 2.22	69.26 ± 34.36	220.81 ± 25.24	112.53 ± 16.95	62.90 ± 20.91	66.86 ± 14.47	37.76 ± 16.56	84.49 ± 33.10
Fe	162.01 ± 23.21	134.55 ± 8.89	65.72 ± 8.65	87.60 * ± 5.39	187.49 ± 19.83	169.83 ± 15.23	83.07 ± 15.47	72.40 ± 18.47
Cu	10.47 ± 1.46	8.51 ± 1.25	13.02 ± 3.82	19.78 ± 3.87	7.90 ± 1.01	9.89 ± 0.98	11.34 ± 2.63	11.15 ± 4.14
Zn	185.05 ± 61.35	177.73 ± 21.59	995.94 ± 69.21	1005.65 ± 101.95	259.17 ± 35.12	281.31 ± 13.03	765.52 ± 404.56	710.54 ± 218.09
Mo	4.47 ± 1.52	5.77 ± 1.62	6.88 ± 1.77	9.92 ± 2.35	136.66 ± 20.36	120.91 ± 25.37	122.94 ± 46.47	95.85 ± 30.37

**Table 2 ijms-22-10166-t002:** Metal contents (Fe, Zn, Cu, Mn, and Mo expressed as μg/g DW) of *Arabidopsis thaliana* seedlings (whole plant) grown under full nutrient conditions (C), absence of iron (-Fe), absence of sulfur (-S), and combined Fe and S deficiency (-Fe-S). Determinations were performed 7 days after transferring seedlings to NNMF conditions Data are shown as the means (±SE) from four independent biological experiments, whereas the asterisk indicates significant (*t* test, *p* < 0.05) differences between NNMF and GMF plants.

	wt		spl7		ami4/5		ami5	
	GMF	NNMF	GMF	NNMF	GMF	NNMF	GMF	NNMF
Mn	94.99 ± 21.41	224.91 ± 45.66	7.93 ± 1.44	16.21 * ± 3.45	60.46 ± 11.78	121.89 * ± 39.85	58.51 ± 15.47	98.27 ± 28.12
Fe	390.01 ± 40.44	317.11 * ± 21.44	184.23 ± 19.84	61.4 * ± 11.74	344.18 ± 58.74	424.28 ± 69.45	188.86 ± 36.75	716.72 * ± 120.56
Cu	11.13 ± 2.12	13.89 ± 1.85	6.2 ± 0.87	19.6 * ± 3.66	20.54 ± 4.83	8.70 * ± 2.14	19.34 ± 4.36	48.33 * ± 19.45
Zn	327.97 ± 59.74	353.90 ± 78.63	448.17 ± 44.62	162.80 ± 63.14	579.30 ± 31.56	252.07 ± 98.12	540.31 ± 65.12	947.73 ± 105.74
Mo	1.35 ± 0.12	8.01 ± 1.87	1.93 ± 0.65	2.00 ± 0.57	4.69 ± 1.63	1.81 ± 0.9	27.36 ± 5.44	7.35 ± 2.85

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
