# Peer review of "Geomagnetic Field (GMF)-Dependent Modulation of Iron-Sulfur Interplay in Arabidopsis thaliana"

_ijms, 2021, doi:10.3390/ijms221810166_

Round 1
Reviewer 1 Report
Vigani et al. ijms-1386884. The authors are presenting a detailed study of the effect of near-null magnetic fields on developmental and molecular parameters of Arabidopsis thaliana. The Turin lab is in the comfortable position to possess excellent equipment, i.e. triple-coil systems, that allow the generation of precision fields even at very low magnetic-flux densities. The techniques are sound and the conclusions drawn are justified. Overall significance: along with previous work from this and other laboratories this work extends and strengthens the notion that the geomagnetic field represents an essential physical parameter that affects plant development on all levels. Accept as is.
Author Response
We thank the Reviewer for the positive evaluation
Reviewer 2 Report
This is a well written and interesting paper reporting on changes in gene expression and on sulphur and iron nutrient uptake, in plants grown in the presence of a near null magnetic field.
In lines 64 to 69 the authors give some background to the effect of the Earth’s geomagnetic field on plant growth. Is the direction and/or angle of the environmental magnetic fields important (vertical in the polar regions and horizontal at the equator) with respect to its influence on plant growth and development?
I am concerned by the reported statistical analysis. The authors state that each of their analytical replicates represented 20 to 30 plants but they then reported only on either 3 replicates (Figs 1 to 5 and Fig S1) or 4 replicates (Tables 1 and 2). The authors need to use more replicates to improve the robustness and significance of their findings.
Minor corrections:
Line 14. . . . nutrient (singular not plural)
Line 34. . . . ability of plants . . (delete definite article)
Line 41-42. . . . being the fourth most abundant element in the Earth's crust after oxygen, silicon, and aluminium.
Line 57. . . . S deficiency inhibits the response to Fe deficiency . . .
Line 97. As expected, the S index strongly. . .
Line 109 Fig 1 legend. . . . 3 independent biological replicates . .
Line 138 Fig 2 legend. . . . three independent experiments . .
Line 167 Fig 3 legend. . . . three independent experiments . .
Line 211 Fig 4 legend. . . . three independent experiments . .
Line 226. Table 2 legend. Determinations were performed . .
Line 226. Table 2 legend. . . . from four independent biological replicates,
Line 238. Fig 5 legend. Root length was measured 7 days . . .
Line 441-442. “The interest in investigating Fe and S interaction relies on the increase of Fe and S dual deficiency in the coming years.” As it stands this sentences needs rewriting. Why is there likely to be a future deficiency in Fe and S mineral nutrients?
Line 443-444. . . . the mechanisms that govern the interconnection of plant Fe and S homeostasis.
Line 445 – 446. “One of the abiotic stress experienced by plants is the variation of the GMF [16].” This sentence needs to be reworded. Plants have NOT been tested for their response to variations in GMF
Line 447. i) near null GMF affects the proper . . . You need to be careful with your use of words – you are comparing growth in two different magnetic fields. The individual plants are NOT being subject to changing magnetic fields.
Fig S1 legend. . . . 3 independent biological replicates . . .
Fig S2 legend. Data are from three independent experiments (N=3) with three replicates for each gene (n=3).
Recommendation:
This paper is currently not suitable for publication unless the authors can provide additional data to improve the robustness of their statistical analysis.

Author Response
This is a well written and interesting paper reporting on changes in gene expression and on sulphur and iron nutrient uptake, in plants grown in the presence of a near null magnetic field.
Authors: we thank the reviewer for the positive evaluation of our manuscript
In lines 64 to 69 the authors give some background to the effect of the Earth’s geomagnetic field on plant growth. Is the direction and/or angle of the environmental magnetic fields important (vertical in the polar regions and horizontal at the equator) with respect to its influence on plant growth and development?
Authors: we thank the Reviewer for the suggestion. We added references indicating that plants respond to both inclination and intensity of the magnetic field.
I am concerned by the reported statistical analysis. The authors state that each of their analytical replicates represented 20 to 30 plants but they then reported only on either 3 replicates (Figs 1 to 5 and Fig S1) or 4 replicates (Tables 1 and 2). The authors need to use more replicates to improve the robustness and significance of their findings.
Authors: We apologize for the low clarity of the text. Following the definition provided by Rogers et al., 2021 (Journal of Experimental Botany, Vol. 72, No. 15 pp. 5270–5274, 2021 doi:10.1093/jxb/erab268), our experimental set-up has been performed by 3 independent biological experiments. Each experiment was organized as follows: 3 Petri dishes (each containing 20 plants) for each treatment (+Fe, -Fe, -S, -Fe-S). Plant sampling was performed by pooling 20-30 representative plants from the 3 Petri dishes. Such set-up has been previously used for other papers (cited in the MS): Islam et al.,,2021, 2020, Agliassa et al., 2018;
Therefore, we modify the text (line 434-436) as follows: “Data are means and (SD o SE) of three (qPCR analysis; n = 3) or four (metals content determination; n = 4) independent experiments each run in triplicate. Each replicate was characterized by pooled tissues (root or shoot) from 20 to 30 plants.”
Minor corrections:
Line 14. . . . nutrient (singular not plural)
Authors: we made the changes accordingly
Line 34. . . . ability of plants . . (delete definite article)
Authors: we made the changes accordingly
Line 41-42. . . . being the fourth most abundant element in the Earth's crust after oxygen, silicon, and aluminium.
Authors: we made the changes accordingly
Line 57. . . . S deficiency inhibits the response to Fe deficiency . .
Authors: we made the changes accordingly
Line 97. As expected, the S index strongly. .
Authors: we made the changes accordingly
Line 109 Fig 1 legend. . . . 3 independent biological replicates .
Authors: we made the changes accordingly
Line 138 Fig 2 legend. . . . three independent experiments .
Authors: we made the changes accordingly
Line 167 Fig 3 legend. . . . three independent experiments .
Authors: we made the changes accordingly
Line 211 Fig 4 legend. . . . three independent experiments . .
Authors: we made the changes accordingly
Line 226. Table 2 legend. Determinations were performed .
Authors: we made the changes accordingly
Line 226. Table 2 legend. . . . from four independent biological replicates,
Authors: we made the changes accordingly
Line 238. Fig 5 legend. Root length was measured 7 days . .
Authors: we made the changes accordingly
Line 441-442. “The interest in investigating Fe and S interaction relies on the increase of Fe and S dual deficiency in the coming years.” As it stands this sentences needs rewriting. Why is there likely to be a future deficiency in Fe and S mineral nutrients?
Authors: we made the changes accordingly. In the conclusion section we added the following sentences: “The interest in investigating Fe and S interaction relies on the expected increase of Fe and S dual deficiency in soil in the coming years. It has been estimated that Fe deficiency impairs plant growth on one third of the cultivated land worldwide and concomitantly, the decrease of S release, related to anthropogenic activities, will increase S deficiency in soils [2]. Therefore, sustaining plant growth and productivity, by optimizing the use of exogenous fertilizers, will necessitate decrypting the mechanisms that govern the interconnection of plant Fe and S homeostasis.”
Line 443-444. . . . the mechanisms that govern the interconnection of plant Fe and S homeostasis.
Line 445 – 446. “One of the abiotic stress experienced by plants is the variation of the GMF [16].” This sentence needs to be reworded. Plants have NOT been tested for their response to variations in GMF
Authors:The sentence has been rephrased
Line 447. i) near null GMF affects the proper . . . You need to be careful with your use of words – you are comparing growth in two different magnetic fields. The individual plants are NOT being subject to changing magnetic fields.
Authors: The sentence has been modified, we deleted “proper”
Fig S1 legend. . . . 3 independent biological replicates . . .
Authors: The sentence has been modified
Fig S2 legend. Data are from three independent experiments (N=3) with three replicates for each gene (n=3).
Authors: The sentence has been modified
Recommendation:
This paper is currently not suitable for publication unless the authors can provide additional data to improve the robustness of their statistical analysis.
Authors: we specified in the text details about the experimental set-up of the work
Round 2
Reviewer 2 Report
I am happy with the author responses and explanation regarding the statistical analysis. I have no further suggestions and recommend publication of the manuscript as is.